# The Role of Medical Image Modalities and AI in the Early Detection, Diagnosis and Grading of Retinal Diseases: A Survey

**DOI:** 10.3390/bioengineering9080366

**Published:** 2022-08-04

**Authors:** Gehad A. Saleh, Nihal M. Batouty, Sayed Haggag, Ahmed Elnakib, Fahmi Khalifa, Fatma Taher, Mohamed Abdelazim Mohamed, Rania Farag, Harpal Sandhu, Ashraf Sewelam, Ayman El-Baz

**Affiliations:** 1Department of Diagnostic and Interventional Radiology, Faculty of Medicine, Mansoura University, Mansoura 35516, Egypt; 2Electronics and Communications Department, Faculty of Engineering, Mansoura University, Mansoura 35516, Egypt; 3Bioengineering Department, University of Louisville, Louisville, KY 40292, USA; 4College of Technological Innovation, Zayed University, Dubai 19282, United Arab Emirates; 5Ophthalmic Department, Faculty of Medicine, Mansoura University, Mansoura 35516, Egypt

**Keywords:** retinal diseases, artificial intelligence, diabetic retinopathy, macular degeneration, modalities

## Abstract

Traditional dilated ophthalmoscopy can reveal diseases, such as age-related macular degeneration (AMD), diabetic retinopathy (DR), diabetic macular edema (DME), retinal tear, epiretinal membrane, macular hole, retinal detachment, retinitis pigmentosa, retinal vein occlusion (RVO), and retinal artery occlusion (RAO). Among these diseases, AMD and DR are the major causes of progressive vision loss, while the latter is recognized as a world-wide epidemic. Advances in retinal imaging have improved the diagnosis and management of DR and AMD. In this review article, we focus on the variable imaging modalities for accurate diagnosis, early detection, and staging of both AMD and DR. In addition, the role of artificial intelligence (AI) in providing automated detection, diagnosis, and staging of these diseases will be surveyed. Furthermore, current works are summarized and discussed. Finally, projected future trends are outlined. The work done on this survey indicates the effective role of AI in the early detection, diagnosis, and staging of DR and/or AMD. In the future, more AI solutions will be presented that hold promise for clinical applications.

## 1. Introduction to Retinal Diseases

Retinal diseases receive serious and widespread attention, as retinopathies are some of the leading causes of severe vision loss and blindness at a global level [1]. Ocular imaging is critical in the management of retinal diseases, especially diabetic eye disease. Advanced imaging modalities allow better understanding of diabetic eye diseases and selection of suitable management options [2].

Multiple retinal diseases can be detected, such as age-related macular degeneration (AMD), diabetic retinopathy (DR), diabetic macular edema (DME), retinal tear, epiretinal membrane, macular hole, retinal detachment, retinitis pigmentosa, retinal vein occlusion (RVO), and retinal artery occlusion (RAO) (see Figure 1). Among these diseases, AMD and DR are the major causes of progressive vision loss, while the latter is recognized as a world-wide epidemic [3]. This review outlines the basic modalities used for the diagnosis of AMD and DR. In addition, the role of AI in diagnosis and staging of these diseases will be surveyed. 

### 1.1. Diabetic Retinopathy (DR)

DR and DME are considered the leading cause of blindness worldwide, and lead to significant visual morbidity [2,4]. The prevalence of DR among diabetic patients may reach 34.6%, while the prevalence of severe DR that threatens the vision is 10.2% [5]. The prevalence of diabetes mellitus (DM) has been continuously increasing over the last three decades due to lifestyle changes [6]. Patients with type 1 DM are more prone to DR than those with type 2 DM. The most important preventable risk factor for DR is hyperglycemia [7].

Screening for DR is a crucial component of DM management [8], thus supported by the fact that major complications of DM that affect vision such as diabetic macular edema (DME) and proliferative DR can respond to treatment [9].

The International Council of Ophthalmology Guidelines for Diabetic Eye Care 2017 recommended that examinations for screening for DR should include retinal evaluation by ophthalmoscope or fundus photography [10]. Early detection of DR through population screening and timely treatment can reduce the retinal complications in diabetic patients and prevent visual loss and blindness [11]. One of the aims of this review is to discuss the different retinal imaging techniques for early detection and grading of DR.

Retinal imaging techniques play a major role in the management and prognosis of diabetic eye disease. Better understanding of the advances in retinal imaging modalities helps in the screening, diagnosis, and treatment of different disease presentations [2,12].

While direct and indirect ophthalmoscopy are the primary techniques for evaluation of DR, several imaging techniques such as color fundus photography (CFP), fundus fluorescence angiography (FFA), ultrasonography, fundus autofluorescence (FAF), and optical coherence tomography (OCT) have proven to be useful depending on the manifestation of the disease [13].

### 1.2. Age-Related Macular Degeneration (AMD)

Macular degeneration or AMD is the primary cause of blindness affecting elder individuals [14]. The progression of the disease involves deterioration with age. Hence, aging is the key factor behind the development and progress of AMD [15]. 

AMD has traditionally been divided into two major types: the non-neovascular (dry or atrophic) form and the neovascular (wet or exudative) form [16]. AMD is also classified into early, intermediate, and advanced AMD, according to the natural course of the disease [17,18]. Non-neovascular AMD represents nearly 90% of all cases and is exemplified by drusen accumulation, the absence of choroidal neovascularization, and retinal pigment epithelium (RPE) atrophy [19]. Neovascular AMD (nAMD) is distinguished by choroidal neovascularization (NV) with abnormal blood vessels that tend to leak fluid or blood [20]. It causes more than 80% of serious vision loss from AMD, with rapid deterioration toward blindness [18]. Untreated nAMD often leads to fibrovascular scarring with associated loss of central visual function [16].

Age at time of diagnosis with AMD is the vital factor in stopping AMD progression. In addition to the aging process, smoking is a risk factor related with AMD. A cohort study observed that smoking doubles the risk of having AMD in 5 years compared to nonsmokers [21]. The precise pathophysiology of AMD is still unknown. Various theories are hypothesized to be the underlying factors for AMD. These include drusen accumulation, chronic inflammation, oxidative stress, reduction of antioxidant, and dysregulated complement [21,22].

Early diagnosis of AMD plays a major role in delaying progression and improving outcomes. Multimodal imaging in diagnosis offers a detailed structure of retinal alteration in AMD without the need for invasive procedures, resulting in early detection and comprehensive management of AMD [6]. Imaging not only plays an important diagnostic role in AMD, but is also used to deliver improved knowledge of its pathophysiology, define treatment options, and assess the treatment response [23]. Imaging helps clinicians to visualize abnormalities, such as RPE atrophy, drusen deposits, subretinal fluid, and choroidal neovascularization [23].

Imaging modalities comprise CFP, FFA, indocyanine green angiography (ICGA), fundus autofluorescence (FAF), OCT, and OCTA [6].

The FFA is still considered by some to be the gold standard for the diagnosis of wet nAMD [5]. However, the concomitant use of FFA with OCT has become the standard in current practice due to the progressive use of OCT as a first-line diagnostic tool for nAMD [7,8].

With the introduction of advanced retinal imaging modalities, the CFP is no longer the primary method for diagnosing and monitoring dry AMD. FAF and OCT are now considered essential methods, whereas other modalities, such as near-infrared autofluorescence (NIA), FFA, and OCTA, may deliver complementary data [24]. 

As shown in Figure 2, imaging modalities are the input of any AI system that aims to detect, diagnose, classify, and/or stage retinal diseases. The goal of this manuscript is to outline the different medical image modalities and technologies that help in the detection, diagnosis, classification, and grading of the retinal diseases, and more specifically, DR and AMD. In addition, this paper aims at providing a review of the literature on AI systems, surveyed from 1995 to 2022, used for the automated detection, diagnosis, classification, and/or staging of DR and AMD.

The rest of this paper is organized as follows. Section 2 summarizes the retinal imaging modalities, their technologies, and their role in the detection and staging of DR and AMD. Section 3 summarizes the noise sources and denoising methods of retinal images. Section 4 introduces the concept of AI to assist the clinicians in the detection and staging processes of retinal diseases. Section 5 and Section 6 specify the role of AI in the detection, diagnosis, and grading of RD and AMD, respectively. Section 7 discusses the findings of the paper and outlines the future trends. Finally, Section 8 concludes the paper.

## 2. Retinal Imaging Modalities

As shown in Figure 2, to build any AI system for the detection, diagnosis, and grading of retinal diseases, the first step is to capture the retinal image using the appropriate medical image modality. Advances in retinal imaging have improved the diagnosis and management of diabetic retinopathy (DR) and age-related macular degeneration (AMD). A summary of the medical image modalities that are used for the detection, diagnosis, and staging of DR and AMD is illustrated in Figure 3. Fundus fluorescein angiography (FFA) is the classic imaging modality for AMD, and is a powerful technology for identifying its presence and degree. Optical coherence tomography (OCT) is now widely used for early diagnosis and determination of the anti-vascular endothelial growth factor therapy (anti-VEGF) retreatment criteria for neovascular lesions. FFA is currently considered the gold standard technique for the evaluation of the retinal vasculature, which is the most affected part of the retina in the diabetic eye. Optical coherence tomography angiography (OCTA) can detect subtle changes in the retinal vasculature before the development of the clinical features of retinopathy, allowing early detection of DR and help in screening for DR among populations at risk. In this section, we will go over the different modalities for the detection and staging of DR and AMD. For each modality, we will illustrate its subtypes and sub-technologies, along with a detailed illustration of how to use this modality for the early detection and staging of both DR and AMD.

### 2.1. Color Fundus Photography (CFP)

Fundus imaging is the process whereby reflected light is used to acquire a two-dimensional image of three-dimensional retinal tissue, with image intensities representing the quantity of reflected light [9]. Fundus photography provides a colored image of the retina. Conventional fundus photography was performed using film, before becoming digitalized. Digital fundus photography has the advantages of rapid acquisition, immediate availability, and the ability to enhance and process the images [10,13].

Types of fundus photography include (i) standard, (ii) widefield/ultra-widefield, and (iii) stereoscopic fundus photography [13].
Standard fundus photography is widely available and easy to use. It captures a 30° to 50° image of the posterior pole of the eye, including the macula and the optic nerve. Standard fundus photography cameras can collect multiple fundus field images. These images are then overlapped to create a montage image with a 75° field of view [10,13].Widefield/ultra-widefield fundus photography can image the peripheral retina. It can capture a 200° field of view even if the pupil is not dilated. This 200° field extends beyond the macula to cover 80% of the total surface of the retina. Theoretically, the large field of view permits better detection of peripheral retinal pathology. However, widefield fundus photography presents some limitations; the spherical shape of the globe causes image distortion, artifacts as a result of eyelashes, and false findings due to inadequate color representation, in addition to the expensive equipment. Consequently, standard 30° fundus photography remains the best choice for fundus imaging [10,13].Stereoscopic fundus photography can be used to obtain a stereo image created by merging photographs taken at two slightly different positions from both eyes to enable the perception of depth [11,13,25]. Despite the potential value of stereoscopic fundus photography, its clinical value is controversial due to several limitations. The acquisition of stereo images is time consuming, and patients must be exposed to double the number of light flashes [11]. The photographer’s experience has an impact on the technique, and the left and right images must be equally sharp and have the same illumination in each image in the pair [12,26]. Image interpretation is time consuming and requires special goggles or optical viewers to fuse the image stereoscopically and achieve depth [11,25].

#### 2.1.1. Application of Color Fundus Photography (CFP) in DR

CFP is widely available, and therefore it is used in screening and clinical trials of DR; it provides good visualization of obvious signs of diabetic macular edema and proliferative diabetic retinopathy such as microaneurysms, liquid exudate, and dot and blot hemorrhages [10]. The wider field of view obtained using the steered images technique in color fundus photography is the basis of the Early Treatment Diabetic Retinopathy Study (ETDR S) grading system, which modifies the Airlie House classification and develops a severity scale of 13 levels [27,28,29]. These levels range from no evidence of retinopathy to significant vitreous hemorrhage [5]. Ultra-widefield color fundus photography allows better detection of peripheral retinal pathology. However, the drawbacks of widefield fundus photography plus the expensive equipment make standard 30° fundus photography the best choice for fundus imaging [10,13].

#### 2.1.2. Application of Color Fundus Photography (CFP) in AMD

CFP is one of the simplest imaging modalities for detecting both dry (non-neovascular) and wet (neovascular) AMD [30]. CFP offers an illustration of variable fundus abnormalities, involving various subtypes of macular drusen and pigmentary abnormalities, and closely parallels biomicroscopic examination [31]. Early funduscopic classification systems of non-neovascular AMD include descriptions of the following: drusen size (i.e., large versus small), consistency (i.e., soft versus hard), location, number, area of involvement, geographic atrophy (GA) size, location, and area [32]. Drusen appears as a yellowish round lesion, with a pigmentary deposit around the macula, while atrophic RPE shows a hypopigmentation around the macula. Meanwhile, the application of CFP in neovascular AMD (nAMD) is helpful in the detection of exudative complications, such as macular edema and macular detachment [33]. CFP has several drawbacks, as the image is created in 2D, and thus lacks proper visualization of small details. Abnormalities in the refractive media, such as cataracts, result in lower image clarity [18]. Additionally, it has a lower sensitivity of 78% when used as an individual imaging procedure to detect choroidal neovascularization, compared to the sensitivity of OCT (94%) [33,34]. CFP alone is deficient for diagnosing nAMD, as it underestimates the presence of choroidal neovascularization [30].

### 2.2. Fundus Fluorescein Angiography (FFA)

FFA is a two-dimensional imaging technique that depends on an intravenous injection of fluorescent dye (resorcinolphthalein sodium) [10,30]. A ring-shaped flash camera is used for excitation of the dye molecules and the projected blue light is reflected from the layers of the retina. Some of the projected light becomes absorbed by the fluorescent dye, and then it is emitted back as green light with a wavelength of 530 nm to be captured by a filter on a digital detector. FFA records the dynamic changes in the retinal and choroidal vasculature [30].

The disadvantages of FFA are general systemic side effects of the injected dye, such as anaphylactic or allergic reaction and nausea, and extensive leak of the dye into the surrounding tissue, which could alter the readings [35].

Types of FFA include (i) indocyanine green angiography (ICGA).
Indocyanine green angiography (ICGA) is a type of FFA based on intravenously injected high-molecular-weight indocyanine green dye. It projects light with a longer wavelength (near infrared light (790 nm)), which allows deeper penetration of the retinal layers, resulting in better visualization of choroidal and retinal circulation [23,36]. Systemic side effects can similarly occur [36]. In ICGA, the dye combines with plasma proteins, leading to less dye leakage than in FFA [37].

#### 2.2.1. Application of FFA in DR

FFA is currently considered the gold standard technique for the evaluation of the retinal vasculature, which is the most affected part of the retina in the diabetic eye [13].

DR signs in FFA images: Microaneurysms: appear as punctate hyperfluorescent areas.Retinal hypoperfusion: nonperfused retinal capillaries, which can cause ischemia and appear as patches of hypofluorescent areas.Increased foveal avascular zone: results from macular ischemia and can explain the cause of loss of vision in some diabetic patients.Retinal neo-vascularization or intraretinal microvascular abnormalities.

Fluorescein dye leaks out from abnormal blood vessels. The visualization of this leak over time is beneficial in the detection of the breakdown of the blood–retinal barrier. Monitoring fluorescein leakage overtime in the macula is very useful in patients with DME. Fluorescein dye leakage also occurs with retinal neovascularization. In proliferative DR patients, this can help in the diagnosis of neovascularization in the optic disc and other areas of the retina [13].

Given the advantage of widefield FFA in imaging of the peripheral retina, it can used in the detection of peripheral retinal neovascularization and the determination of the extent of peripheral areas of capillary nonperfusion and hypoperfusion [38,39].

#### 2.2.2. Application of FFA in AMD

FFA is the gold standard for nAMD, compared to other modalities. FFA outperforms its rivals in specifying choroidal neovascularization (CNV) in its structural and leakage state. Based on the location of CNV, it can be classified as extrafoveal, subfoveal, and juxtafoveal [30,40]. Recognition of the CNV location is a valuable prognostic factor. Extrafoveal CNV is situated around 200–2500 μm from the center of the foveal avascular zone (FAZ), subfoveal CNV is situated beneath the center of the FAZ, while juxtafoveal CNV is situated up to 199 μm from the center of the FAZ and some part of the FAZ excluding the center portion [30].

FFA also indicates the leakage properties of the CNV, which can be categorized into occult CNV (type I), classic CNV (type II), and retinal angiomatous proliferation (type III) [23]. Occult CNV exists as mottled and patchy hyperfluorescence in early-phase angiograms and leaks in the later phase, forming larger hyperfluorescent dots [30]. The occult CNV is subgrouped into two types on the basis of its leakage features. Type I occult CNV is fibrovascular and is defined as stippled hyperfluorescence in the early phase, with progressive leakage upon late-phase angiogram. Meanwhile Type II occult CNV consists of late leakage from an unspecified source and does not appear in the early phase, but displays speckled hyperfluorescence upon mid- to late-phase angiogram [30,41]. Classic CNV presents as a well-defined hyperfluorescence network membrane in the early phase, followed by progressive leakage upon late-phase angiogram [42]. Retinal angiomatous proliferation (RAP) was recently described as NV arising from the intraretinal layer and infiltrating into the choroid layer, forming a retinal–choroidal anastomosis. RAP can be divided into three stages based on the extent of NV and proliferation [43].

Disadvantages of the FFA procedure include its systemic complication from the injected dye. Additionally, the dye leaks considerably to the surrounding tissue, which might affect the detailing of the CNV. Hence, in some cases of type I CNV and RAP, ICGA is a preferable method to FFA [44].

#### 2.2.3. Application of Indocyanine Green Angiography (ICGA) in AMD

ICGA uses a high-molecular-weight contrast that binds to plasma proteins and thus leaks less compared to FFA [45]. ICGA is well established in the detection of type I CNV and occult CNV: the early phase often shows ill-defined hypercyanescent lesions, the mid phase shows progressive intensity, and the late phase exhibits hypercyanescent plaque. However, ICGA is less appropriate for detecting classic CNV, which appears as well-defined hypercyanescence [46].

Polypoidal choroidal vasculopathy (PCV) is an abnormal choroidal vascular network with aneurysmal dilatation (polypoidal characteristics) at its periphery [24]. On ICGA, PCV exists as a hypercyanescent hot spot in early angiograms, with a grape-like/polypoid structure. PCV frequently masks the appearance of occult or classic CNV on FFA, and therefore ICGA is considered the gold standard in the detection of PCV, as its appearance is masked by the RPE layers in FFA [30]. PCV is associated with neurosensory detachment, high numbers of recurrent cases, and poor visual outcomes [31].

RAP is best visualized by ICGA and appears in the early phase as a hyperfluorescent hot spot with apparent retinal artery communication into the CNV, followed by a progressive increase in both size and intensity in the late phase [43]. The recognition of RAP on one eye is associated with increased incidence of NV on the other eye, with nearly a 100% risk in 3 years of follow-up [47].

ICGA uses a longer wavelength of infrared compared to FFA, thus allowing deeper penetration into the RPE, choroidal structure, and any subretinal fluid, hemorrhages, or pigment epithelium detachment, which often alter imaging in FFA. Hence, in the presence of hemorrhage, ICGA images offer a more detailed overview of the characteristics of CNV in AMD patients compared to FFA [30].

### 2.3. Fundus Autofluorescence Imaging (FAF)

FAF is a noninvasive imaging technique for the mapping of natural or pathological fluorophores of the ocular fundus. The dominant source of fluorophores is the lipofuscin located in the retinal pigment epithelium; lipofuscin is responsible for the fluorescent properties necessary for FAF imaging [48]. A light with a specific wavelength of 300–500 nm is used to stimulate fundus fluorescent properties without the use of contrast material, and excites the lipofuscin particles, which then emit a light with a wavelength of 500–700 nm [18,33,37]. 

FAF can be performed using a fundus camera, fundus spectrophotometer, or confocal scanning laser ophthalmoscope. The best choice is confocal scanning laser ophthalmoscope, because of its ability to decrease the noise from other autofluorescence materials from the anterior eye segment [33,49].

Types of fundus FAF include near infrared autofluorescence (NIA).
Near infrared autofluorescence (NIA) is another fundus imaging technique that uses the other fluorophore properties of the retina located in melanin. Melanin is present mainly in the retinal pigment epithelium, and to a lesser extent in the choroid in small amounts. NIA uses diode laser light with a longer wavelength of 787 nm for excitation, and then a specific wavelength above 800 nm is captured using a confocal scanning laser ophthalmoscope [50,51]. The captured image shows increased hyperautofluorescence in the center of the fovea due to the high melanin content of the retinal pigment epithelial cells [50]. Retromode imaging (RM) is an imaging modality using an infrared laser at 790 nm, generating a pseudo-3D appearance of the deeper retinal layer [52].

FAF techniques include (i) fundus spectrophotometry, (ii) scanning laser ophthalmoscopy, (iii) fundus camera, and (iv) widefield imaging.
Fundus spectrophotometry is able to process the excitation and emission spectra of autofluorescence signals originating from a small retinal area of the fundus (only 2° in diameter) [53]. It is composed of an image intensifier, diode array detector, and crystalline lens. The beam is separated in the pupil, and the detection is confocal to reduce the contribution of the crystalline lens in the autofluorescence. The complex instrumentation and the small examined area have led fundus spectrophotometry not to be the preferred technique in clinical practice for FAF [48,53].Scanning laser ophthalmoscopy can image larger areas of the retina by using a low-power laser beam that is projected onto the retina and distributed over the fundus [54]. Then, the reflected light intensities from each point after passing through a confocal pinhole are collected via a detector, and the image is produced [48]. A series of several images are recorded, then averaged to form the final image, reduce the background noise, and improve the image contrast [55].Fundus cameras have limitations with respect to FAF, such as weak signal, the crystalline lens absorptive effect, nonconfocal imaging, and light scattering [48]. A modified fundus camera was designed by adding an aperture to the illumination optics to decrease the effect of light scattering from the crystalline lens and reduce the loss of contrast [56]. This modified design is limited by the small field of view (only 13° in diameter) and complex instrumentation [48].Widefield imaging: confocal scanning laser ophthalmoscopy has a 30° × 30° retinal field. Therefore, imaging of larger retinal areas like a 55° field needs additional lenses. The fundus camera can be used to manually produce montage images using seven field panorama-based software packages [48].Widefield scanning laser ophthalmoscopy was developed to record peripheral autofluorescence images using green light excitation (532 nm) with an acquisition time of less than two seconds. The widefield extends beyond the vascular arcades and can be used to assess the peripheral involvement of retinal diseases [48]. Ultra-widefield scanning laser ophthalmoscopy was developed by combining confocal scanning laser ophthalmoscopy with a concave elliptical mirror. It can record a wider view of the retina of up to 200° in a single image with an acquisition time of less than one second, without the need for pupil dilatation [25,57]. The use of ultra-widefield scanning laser ophthalmoscopy is still limited due to its high cost [12].

#### 2.3.1. Application of FAF in DR

Previous studies have reported increased autofluorescence in patients with DME [58,59]. Multiple patterns have been used to describe the FAF findings: single cyst, multiple cysts of increased FAF, or both combined [60]. Other patterns include normal, increased FAF, single spot, and multiple spots of increased FAF [61].

In DME, an association has been reported between increased FAF and decreased visual acuity [59]. Follow-up visits for patients with DME revealed that patients with deteriorated vision had increased FAF compared to patients with stationary or improved vision [2].

#### 2.3.2. Application of FAF in AMD

FAF is an imaging method that uses a specific wavelength of light to trigger the fundus fluorescence characteristics without the need for contrast [33]. FAF images have the ability to detect numerous retinal abnormalities, such as pigmentary changes, drusen, geographic atrophy, and reticular pseudodrusen [49].

Drusen exists in numerous patterns on FAF, such as hypoautofluorescence, hyperautofluorescence, and normal lesions, depending on the variability of the fluorophore contents and the size of the drusen [49,62]. FAF is the gold standard modality for the assessment of GA, as it offers high-contrast retinal images that can be used to detect areas of atrophy. Atrophic lesions present as hypoautofluorescent areas, owing to the loss of the RPE cells containing intrinsic fluorophores, such as lipofuscin [16]. The disparity between areas of RPE loss and adjacent areas of intact photoreceptors allows the reproducible semiautomated quantification of atrophic areas. Therefore, FAF has been recognized as an anatomic outcome parameter for the progression of GA in clinical trials by international agencies [45,63].

Patchy, linear, or reticular patterns recognized on FAF have been associated with the development of nAMD, while the patchy pattern is the highest-risk FAF pattern for conversion to nAMD [64,65]. Hemorrhages, scarring, and fibrovascular membranes are hypoautofluorescence lesions, while subretinal fluid appears hyperautofluorescent [49].

The currently most frequently used FAF imaging method uses a confocal scanning laser ophthalmoscope (cSLO) with a blue light excitation wavelength filter (488 nm) and an emission filter of 500 to 521 nm. In comparison with CFP, FAF has the capacity to detect retinal changes in early and intermediate AMD that may appear normal in CFP [50].

FAF has high sensitivity in identifying nAMD (93%), but relatively low specificity (37%) compared to FFA as the gold standard [65]. Obstacles to FAF imaging comprise susceptibility to media opacities, difficult foveal imaging due to macular pigment that absorbs blue light, and patient discomfort [66]. Alternate wavelengths, such as green light, have advantages. For example, it may be more comfortable for patients, and it can reduce macular pigment absorption. However, it can still generate an excellent visualization of the atrophic areas [44].

NIA employs the other fluorophore properties of the retina and melanin [46]. The NIA images show high hyperautofluorescence in the center of the fovea due to the elevated melanin content in RPE cells [50].

Both NIA and FAF appear dark in the atrophic region in dry AMD, while the adjacent area appears to possess increased intensity. Half of AMD patients had increased NIA at the normal FAF site, thus suggesting that there is an increase in melanin activity preceding lipofuscin activity [60]. In nAMD, the image seems dark in both NIA and FAF owing to the blockage of the autofluorescence signal by subretinal fluid, hemorrhage, or choroidal NV [30,67]. Nevertheless, FAF (56.5%) is more efficient at describing exudative activity than NIA (33.9%) [51].

RM is helpful for distinguishing pathological structures in dry and wet AMD. For example, drusen is more obvious in RM compared to in fundus photography [52]. In wet AMD, RM has a higher agreement with OCT in imagining macular edema, but relatively low for RPE detachment [68].

### 2.4. Optical Coherence Tomography (OCT)

OCT plays a crucial role in the diagnosis and management of retinal diseases, as it provides detailed cross-sectional images of the retina, so that ophthalmologists can detect changes in anatomy and monitor treatment response [2]. 

OCT uses light waves to generate the image in a method comparable to ultrasonography, using reflected light, instead of sound, to create the image. Low-coherence light is scanned and concentrated on the ocular structure of interest using an internal lens. A second beam internal to the OCT unit is used as a reference, and a signal is formed by calculating the variation between the reference beam and the reflected beam. Detection of these beams depends on the time-domain or spectral-domain protocols [69].

OCT is the most powerful diagnostic tool for retinal diseases due to its noninvasive, unique, and high-resolution evaluation of tissue, with direct correspondence to the histological appearance of the retina, achieving axial resolution of up to 2–3 µm in tissue. OCT has other advantages that involve reproducibility, noninvasiveness, and repeatability. Additionally, OCT is obtainable across most media opacities, including vitreous hemorrhage, cataract, and silicone oil.

OCT provides a superior, noninvasive modality for evaluating DME [2]. In addition, the spectral domain (SD)-OCT is the gold standard for the most important macular diseases [70]. The introduction of OCT has altered the clinical management of several retinal diseases, involving AMD [71], DME [72], and RVO [73,74].

OCT technologies include (i) time-domain OCT (TD-OCT), (ii) spectral-domain OCT (SD-OCT), (iii) swept-source OCT (SS-OCT), (iv) high-speed ultra-high-resolution OCT, (v) optical coherence tomography angiography (OCTA), (vi) intraoperative optical coherence tomography, and (vii) functional optical coherence tomography.
TD-OCT is the first commercially offered OCT device based on time-domain detection that shows rather low scan rates of 400 A-scans per second. The key imitations in the clinical use of TD-OCT are the limited resolution and slow acquisition [75]. However, it is commonly accepted for the evaluation of several retinal diseases, such as macular edema, AMD, and glaucoma [76].Spectral domain OCT (SD-OCT): Subsequently, spectral domain imaging technologies have significantly improved sampling speed and signal-to-noise ratio by using a high-speed spectrometer that measures the light interferences from all time delays simultaneously [77]. In commercially available SD-OCT devices, technical improvements have enabled scan rates of up to 250,000 Hz [78]. SD-OCT’s higher acquisition speeds allow for a shift from two-dimensional to three-dimensional images of ocular anatomy. In addition, SD-OCT is several orders of magnitude more sensitive than TD-OCT [75]. SD-OCT is used to diagnose DR and diabetic macular edema (DME).SS-OCT technology has also improved imaging accuracy by using a swept laser light source that successively emits various frequencies in time and photodetectors to measure the interference [79]. SS-OCT devices employ a longer wavelength (>1050 nm) laser light source and have scan rates as fast as 200,000 Hz. The longer wavelengths are thought to enhance visualization of subretinal tissue and choroidal structures [80,81]. SS-OCT has been used to visualize a thick posterior hyaloids among eyes with diabetes compared to normal controls [82]. SS OCT can be used to reveal adhesion between the retina and detached posterior hyaloid in eyes with DR and DME, while this was not detected in eyes without diabetic eye disease [2].High-speed ultra-high-resolution OCT (hsUHR-OCT) is another variation on SS-CT that provides a striking improvement in terms of cross-sectional image resolution and acquisition speed. The axial resolution of hsUHR-OCT is approximately 3.5 µm, compared with the 10 µm resolution in the standard OCT. This enables superior visualization of retinal morphology in retinal abnormalities. The imaging speed is approximately 75 times faster than that with standard SD-OCT. hsUHR-OCT improves visualization by obtaining high-transverse-pixel density and high-definition images [83,84].OCTA is a relatively new modality for visualizing flow in the retinal and choroidal vasculature. Rapid scanning by SD-OCT or SS-OCT devices allows analysis of variation of reflectivity from retinal blood vessels, permitting the creation of microvascular flow maps. This technology enables clinicians to visualize the microvasculature without the need for an intravenous injection of fluorescein [2]. OCTA signifies progression of OCT technology, as motion contrast is used to create high-resolution, volumetric, angiographic flow images in a few minutes [85]. Neovascularization at the optic disc is obviously visualized on OCTA, and microaneurysms exist as focally distended saccular or fusiform capillaries on OCTA [86].Intraoperative optical coherence tomography: Performing intraoperative OCT in the operating theater may offer supplementary data on retinal structures that were inaccessible preoperatively due to media opacity [2]. Prospective intraoperative and perioperative ophthalmic imaging with OCT study has been performed to assess the feasibility, utility, and safety of using intraoperative OCT through different vitreoretinal surgical procedures. The information achieved from intraoperative OCT permit surgeons to evaluate subtle details from a perspective distinctive from that of standard en face visualization, which can improve surgical decisions and patient outcome [87]. Intraoperative OCT revealed variable retinal abnormalities in patients who underwent pars plana vitrectomy for dense vitreous hemorrhage secondary to DR, including epiretinal membranes (60.9%), macular edema (60.9%) and retinal detachment (4.3%). The surgeons reported that intraoperative OCT impacts their surgical decision making, particularly when membrane peeling is accomplished [88].Functional OCT makes it possible to perform noninvasive physiological evaluation of retinal tissue, with respect to factors such as its metabolism [89,90]. A transient intrinsic optical signal (IOS) is noted in retinal photoreceptors implying a distinctive biomarker for ocular disease detection. By developing high spatiotemporal resolution, OCT and using an algorithm for IOS processing, transient IOS could be recorded [89]. IOS imaging is a promising alternative for the measurement of retinal physiological functions [91]. Functional OCT provides a noninvasive method for the early disease detection and improved treatment of retinal diseases that cuase changes to retinal function and photoreceptor damage, such as DR and AMD, which can be detected using functional OCT as differences in IOS [2,89].

Functional extensions to OCT increase its clinical potential. For example, polarization-sensitive OCT (PS-OCT) delivers intrinsic, tissue-specific contrast of birefringent (e.g., retinal nerve fiber layer (RNFL)) and depolarizing (e.g., retinal pigment epithelium (RPE)) tissue with the use of polarized light. This allows PS-OCT to be helpful for the diagnosis of RPE disorders in some disease such as AMD [92].

Another extension is Doppler tomography, which allows depth-resolved imaging of flow by observing differences in phase between successive depth scans. This technology offers important data about blood flow patterns in the retina and choroid, granting absolute quantification of the flow within retinal vessels [93].

#### 2.4.1. Application of OCT in DR

OCT has become the gold standard method for the evaluation and management of DME by visualizing changes in the retinal anatomy caused by DME and monitoring the response to treatment [2,10]. OCT is able to determine whether DME is center involving or noncenter involving, which affects the therapy plan [13]. 

DME causes several morphologic patterns; diffuse thickening of the retina, intra-retinal cystic spaces, vitreofoveal traction with loss of the foveal depression, and loss of the external limiting membrane. These patterns are correlated with the degree of visual impairment and the thickness of the retina [10,94,95]. In severe forms of DME, subretinal fluid and focal retinal detachment occur, appearing as voids or dark spaces between the retina and retinal pigment epithelium [10]. Hard exudate on OCT appears as hyperreflective punctate foci in the outer plexiform layer. As well as hemorrhage in different layers of the retina, cotton wool spots in the superficial layer of the retina may be visualized. The appearance of retinal neovascularization on OCT is a highly reflective spot on the inner surface of the retina [10]. Hyaloid traction and preretinal membranes cause distortion of the retinal architecture on OCT, and identification of this traction is crucial for the management of DME, as eyes will not respond to treatment and may require surgery for the resolution of DME. OCT cannot diagnose macular ischemia like FFA does, which in turn limits the ability of OCT to be correlated with anatomical changes with visual acuity [10].

There are multiple classifications for DME based on OCT findings. Patterns are described as diffuse or sponge-like retinal thickening, cystoid macular edema, serous subretinal fluid without posterior hyaloid traction, with posterior hyaloid traction, and mixed patterns [96]. Identification of these different patterns directly affects the diagnosis and treatment of DME [10].

#### 2.4.2. Application of OCTA in DR

OCTA can be used to visualize abnormal flow patterns or irregular vessel geometries in DR, in order to diagnose retinal neovascularization, capillary nonperfusion and microaneurysms. Studies confirmed that OCTA can detect subtle changes in the retinal vasculature before the development of the clinical features of retinopathy allowing early detection of DR and help in the screening for DR among populations at risk [85].

With regard to the visualization of microaneurysms, OCTA can be used to detect the intra-retinal depth of extension of microaneurysms, although it appears less sensitive than FFA for the detection of microaneurysms [97]. Microaneurysms in OCTA appear as dilated capillary segments or loops, small neovascularization foci, or focal capillary dilatations in area adjacent to the capillary nonperfusion [97]. OCTA can create quantitative measurements of the avascular zone of the fovea, capillary nonperfusion areas, flow maps, and vessel density analysis. These quantitative data can provide more detailed and precise information than that obtained using FFA [98].

#### 2.4.3. Application of OCT in AMD

OCT is one of the most suitable noninvasive imaging modalities for identifying and monitoring AMD. There are four hyperreflective bands detected in AMD patients using OCT, which are assumed to represent the external limiting membrane, the inner/outer segment of the photoreceptor, RPE, and Bruch’s membrane [99]. OCT is able to demonstrate AMD abnormalities such as drusen deposits, pseudodrusen, subretinal fluid, RPE detachment, and choroid NV. Drusen deposits present as low mounds underneath the RPE layer, while pseudodrusen presents as a hyperreflective deposit located beneath the retina layer [23]. 

The existence of pseudodrusen in AMD patients is associated with increased risk of GA or nAMD. OCT has the highest sensitivity and specificity for the detection of pseudodrusen among all of the other imaging modalities [100]. OCT is frequently used as a reference imaging method for evaluating the response of nAMD to anti-vascular endothelial growth factor therapy [101,102]. A recent study demonstrated that SD-OCT or FA combined with CFP had similar sensitivity and specificity, with no statistical difference for the primary diagnosis of NV secondary to AMD [103]. 

In GA, the RPE atrophy shows a feathered-like form projected deep into the RPE [36]. OCT additionally displays a progressive loss of retinal bands, which is related to the external limiting membrane, the inner/outer segments of the photoreceptor layer, the RPE layer, and the outer nuclear layer [33]. The enlargement of the atrophic region is linked with the gradual loss of the outer hyperreflective bands and the thinning of the outer nuclear layer, the outer plexiform layer, and the RPE membrane during 12 months of follow-up. Additionally, GA is related to a 14.09 μm increase in retinal thickness [104].

NV activity is assessed on OCT based on the accumulation of fluid at different levels of the retina. Subretinal fluid is depicted as a hyporeflective lesion situated above the RPE and below the retina [23]. RPE detachment looks like a dome shape on the RPE layer [36]. Exudative activity is one of the defining factors for nAMD treatment; increased choroid thickness might represent a possible choroid, but cannot distinguish between classic AMD and PCV [23]. Outer retinal tubules are another structural retinal abnormality in OCT that appears as a hyporeflective center with a hyperreflective border. It represents the degenerated photoreceptors, and thus does not represent exudative activity for NV and does not require treatment for nAMD [105]. 

OCT has drawbacks with respect to grading choroidal NV; however, these can be overwhelmed by performing FFA or OCTA in combination with OCT when indicated [30].

#### 2.4.4. Application of OCTA in AMD

In contrast to both FFA and ICGA, OCTA is a noninvasive procedure for retinal vascular imaging, and has rapidly achieved acceptance for the detection and monitoring of nAMD [106]. The presence of a choroidal NV on OCTA links perfectly with findings on structural OCT and FFA [107]. The improved definition of NV on OCTA has led to an improved understanding of the structural evolution of these lesions with anti-angiogenic treatment [108]. Despite inactivity on FFA, a vascular network can remain persistent on OCTA [109].

OCTA has equivalent detection ability with respect to visualizing CNV to that of FFA and ICGA [30]. In nonexudative CNV, OCTA is valuable in visualizing choriocapillaris blood flow with a significant decrease in choriocapillaris flow in atrophic zone reaching outside the GA area in dry AMD [110]. OCTA revealed a significantly decreased vessel density—by 9%—in dry AMD patients in both superficial and deep vascular layers compared with healthy individuals [111]. 

Nonexudative CNV is often identified by OCTA in the eyes of patients with exudative CNV, with a high risk of exudation developing within the first year after detection. Those patients could benefit from close monitoring [84].

CNV is detected as a hyperfluorescent high flow network with variable depth of the retina involvement according to the degree of CNV [23]. Type I CNV is emerging as a minimal delineated vascularization developing from choriocapillaris and RPE penetrating the Brusch’s membrane, but does not penetrate the RPE layer with no evidence of NV in the outer retina [112,113]. Type II CNV seems like a sharp demarcated vascular change at the choroid, choriocapillaris, and RPE, and extending to the outer retina [113]. Type III CNV is emerging as a hyperreflective cluster located in outer retinal layer with interconnecting vessels and inner retinal circulation [106].

OCTA has a number of limitations, as subretinal hemorrhage diminishes its signal to detect CNV. In addition, OCTA has lower sensitivity compared to FFA for the detection of exudative AMD in cases with large subretinal hemorrhages [114]. Furthermore, OCTA may underestimate the CNV size compared to ICGA [115].

### 2.5. Adaptive Optics (AO)

AO is an adapted technology in which scanning laser ophthalmoscopy (SLO) and OCT are employed to resolve optical aberrations on the basis of retinal imaging. AO grants noninvasive visualization and quantification of retinal capillaries, as it can deliver high-resolution images of the foveal cones, dynamic images of the retinal vasculature, and calculate arterial wall measurements and blood flow speed [116,117]. However, AO is limited by its very small field of view—in the range of 1–2 degrees—which hinders its clinical benefit.

#### 2.5.1. Application of AO in DR

In diabetic eyes, AO has been used to show the irregular branching of blood vessels, shunt vessels and narrowed perifoveal capillaries [118,119]. The diminished regularity of the cone photoreceptor arrangement determined with AO-SLO has been correlated with increasing DR severity and DME [120]. Additionally, an association between capillary nonperfusion in the deep capillary plexus and abnormalities in the photoreceptor layer in DR has been reported using both AO-SLO and OCTA [121].

#### 2.5.2. Application of AO in AMD

AO promotes the correction of ocular aberrations, increases lateral resolution, and decreases artifacts. AO-OCT enhances the ability of OCT to grant early recognition of cellular pathology before visual changes occur [122]. AO-OCT reveals higher reflectivity and reduced speckle size in AMD compared to in OCT [123]. In GA, AO-OCT reveals detailed membrane loss, inner and outer segment loss, and RPE loss. Additionally, in advanced GA, the AO-OCT detects calcified drusen and drusenoid pigment epithelial detachment, thus allowing direct visualization of the photoreceptor destruction caused by drusen [63].

### 2.6. Ultrasound Imaging

Ophthalmic B-scan ultrasound is a rapid, noninvasive imaging technique that creates real-time high-resolution images of the eye with minimum discomfort [124]. 

#### Application of Ultrasonic Imaging in DR

In DR, B-scan ultrasound imaging can determine the status of the retina when visibility is obscured by hemorrhage or dense cataract. It can illustrate the causes of low vision in patients with DR, such as vitreous hemorrhage, asteroid hyalosis, and retinal detachment, with a better assessment of the complications that predict the visual outcome [125]. Ophthalmic ultrasound can also accurately illustrate ocular emergencies, such as retinal detachment and ocular trauma [126]. Additionally, ophthalmic ultrasound is very helpful for relieving the risk of vision loss associated with central retinal artery occlusion [127]. B-scan ultrasound imaging is not sensitive enough to evaluate for DME, and it has a restricted efficiency when the ocular media is clear [128].

## 3. Denoising of Retinal Images

To process retinal images using AI, denoising represents a preprocessing step that may help to improve the AI efficiency for the detection, diagnosis, and staging of retinal diseases. Noise sources in CFP include additive and multiplicative noise [129]. For OCT, noise sources include speckle noise, shot noise, and additive white Gaussian noise (AWGN) [130]. For OCTA, noise sources include speckle noise and AWGN [131]. For FFA, noise sources include the internal noise of sensitive components, optical material grain noise, thermal noise, transmission channel interference, and quantization noise [132]. Most popular denoising techniques include using a Gaussian filter, a median filter, a wavelet filter, and/or a spatial domain filter. More recently, deep autoencoders play a significant role in image denoising. However, recent directions of deep learning attempt to efficiently train the deep learning network to process directly noisy images without the need for preprocessing techniques.

## 4. The Role of AI in the Diagnosis of Retinal Diseases

Artificial intelligence (AI) is a field of knowledge that refers to the imitation of the way in which humans think and solve problems using artificially intelligent components. Machine learning is a basic part of AI. Machine learning depends on extracting features from the input database using different image processing tools, and either categorizing the data based on unsupervised learning or classifying the data into grades using supervised learning. Supervised learning refers to data classification based on supervision (i.e., through labeled input–output pairs; each pair contains an input associated with its desired ground truth output). Classifiers include supported vector machines (SVM), random forest, traditional neural networks (neural networks that are composed of two layers, where the traditional back propagation algorithm is used to adjust the weights), see Figure 4. Recently, deep learning, which is a part of machine learning, has gained a lot of popularity and potential applications in the medical field. The most popular deep learning networks are convolutional neural networks (CNNs). Unlike traditional neural networks, CNNs are composed of many convolutional and fully connected layers that perform both feature extraction and classification. On the other hand, unsupervised learning does not depend on supervision (labeled input output pairs) to perform data categorization. Instead, the patterns of the input data are used to efficiently categorize the data.

Nowadays, AI plays a major role in many applications, including in the detection, diagnosis, grading, and classification of eye diseases (see Figure 2). In this paper, we will briefly survey the different AI-based methods for the early detection, diagnosis, grading, and classification of eye diseases. We will focus on the two major eye diseases: DR and AMD. 

To measure the performance of AI components, different metrics are used to solve medical problems, such as the early detection, diagnosis, and classification of eye diseases. In this section, we will provide a brief overview of these metrics.

### Performance Metrics

Let *TP* indicate true positive, *TN* indicate true negative, *FN* denote false negative, and *FP* denote false positive. The following performance metrics are defined as follows:Specificity:
$$Spef=\frac{Number\;of\;true\;postive\;assesments}{Number\;of\;all\;postive\;assesments}=\frac{TP}{TP+FN}$$Sensitivity (recall):
$$Sen=\frac{Number\;of\;true\;negative\;assesments}{Number\;of\;all\;negative\;assesments}=\frac{TN}{TN+FP}$$Accuracy:
$$ACC=\frac{Number\;of\;correct\;assesments}{Number\;of\;all\;assesments}=\frac{TP+TN}{TP+TN+FP+FN}$$*F*1-score:
$$F1=\frac{TP}{TP+0.5\left(FP+FN\right)}$$Precision:
$$Pre=\frac{Number\;of\;true\;postive\;assesments}{Number\;of\;all\;postive\;assesments}=\frac{TP}{TP+FP}$$Kappa:

$$k=\frac{p_{o}-p_{e}}{1-p_{e}}$$
where $p_{o}=\frac{Number\;of\;Agreements\;among\;raters}{Total}$ and $p_{e}$ is the hypothetical probability of chance agreement.
AUC is the area under the curve of the receiver operating characteristics (ROC), a curve that relates the false positive rate (specificity, on the x-axis) to the true positive rate (sensitivity, on the y-axis). AUC is between 0 and 1. The closer the AUC to 1, the better the performance.Confusion matrix, which is a summary of classification results based on highlighting the number of correct and incorrect predictions for each class.

## 5. The Role of AI in the Early Detection, Diagnosis, and Grading of DR

DR is an epidemic disease [133,134]. In this section, the automated methods for the detection, diagnosis, and staging of DR are outlined.

### 5.1. Traditional Machine Learning Methods

Traditional machine learning (ML) methods involve extracting features from input data using different image processing tools and using a separate classifier to perform classification. These methods may include a feature selection and reduction algorithm to select the most relevant features to the specific classification problem. In the literature, different ML methods have been applied for the purpose of the detection, diagnosis, and grading of DR. These methods are different with respect to the image modality used, the feature extracted, and the classifier used. The most used image modality is fundus imaging (10 out of 18 research studies), followed by OCT, and then OCTA. Please note that the OCT and OCTA modalities have more recently become the modalities of choice (i.e., between 2020 and 2022). Features include statistical features, texture feature and morphological (shape) features. The most used classifiers are the SVM and traditional neural networks. Figure 5 summarizes the traditional ML methods used for the job of detecting, diagnossing, and grading DR.

For fundus images, different features obtained using different image processing algorithms have been used. For example, Welikala et al. [135] used local morphology features with a genetic feature selection algorithm to select the most relevant features for the detection of new vessels from fundus images as an indication of PDR. The detection was performed using an SVM classifier. Prasad et al. [136] used 41 statistical and texture features followed by a Haar wavelet transform for feature selection and principal component analysis (PCA) for feature reduction. A back propagation neural network and one rule classifier were used for the detection of DR from fundus images. Mahendran et al. [137] used both statistical and texture features extracted using a gray-level co-occurrence matrix (GLCM) applied on segmented fundus images. They used SVM and neural networks to detect abnormal DR and then to classify abnormal DR into moderate NPDR or severe NPDR. Bhatkar et al. [138] used discrete cosine transform and statistical features to detect DR using fundus images. A multi-layer perceptron neural network was used for the discrimination of abnormal DR images from normal ones. Labhade et al. [139] classified the data into four classes: normal, mild NPDR, severe NPDR, and PDR using 40 statistical and GLCM texture features extracted from fundus images. Different classifiers have been investigated, including SVM, random forest, gradient boost, AdaBoost, and Gaussian naive Bayes, with the SVM classifier achieving the best performance. Rahim et al. [140] classified fundus images into five classes: no DR, mild NPDR, moderate NPDR, severe NPDR, and PDR. Three features were used: area, mean, and standard deviation of two extracted regions (i.e., retina and exudates), which were segmented using fuzzy techniques. An SVM with a radial basis function (RBF) kernel was used for classification. Islam et al. [141] discriminated between normal and DR fundus images using sped up robust features, followed by k-means, a bag of words approach, and SVM classifiers. Carrera et al. [142] classified nonproliferative DR into four grades using fundus images. They extracted features from isolated blood vessels, microaneurysms, and hard exudates, and an SVM was used to perform classification. Somasundaram and Alli [143] differentiated between NPDR and PDR. They extracted the candidate objects (blood vessels, optic nerve, neural tissue, neuroretinal rim, optic disc size, thickness and variance), and a bagging ensemble was used for classification. Costa et al. [144] graded DR using fundus images. They used a weakly supervised multiple instances learning framework based on joint optimization of the instance encoding and the image classification stages.

For OCT images, different methods have been applied. For example, Sharafeldeen et al. [145] detected DR from OCT images using features that were extracted from 12 retinal layers, including the thickness, tortuosity, and reflectivity of each layer. Two-level neural networks were used for classification. Wang et al. [146] extracted foveal avascular zone (FAZ) metrics, vessel density, extrafoveal avascular area, and vessel morphology metrics from OCT images. A multivariate regression analysis was used to identify the most discriminative features for grading DR. Abdelsalam et al. [147] used multifractal geometry and an SVM for early diagnosis of NPDR using OCTA. Elsharkawy et al. [148] applied majority voting on an ensemble of neural networks, where the input of the NN was the Gibbs energies extracted from the 12 layers of the retina. Table 1 summarizes the different traditional machine learning methods that have been used for DR detection, diagnosis, and grading since 2015.

For OCTA images, different techniques have been employed. For example, Eladawi et al. [149] achieved early detection of DR using OCTA. They extracted features like the density and appearance of the retinal blood vessels, and the distance map of the foveal avascular zone, and an SVM was used for classification. Alam et al. [150] achieved early detection of DR using OCTA images. Features are extracted including blood vessel tortuosity, blood vascular caliber, vessel perimeter index, blood vessel density, foveal avascular zone area, and foveal avascular zone contour irregularity; then, an SVM was used for classification. Liu et al. [151] detected DR using OCTA. A discrete wavelet transform was applied to extract texture features from each image. Different numbers of classifiers were investigated, including logistic regression, logistic regression regularized with the elastic net penalty, SVM, and the gradient boosting tree.

Mixed modalities have also been investigated. For example, Sandhu et al. [152] diagnosed NPDR using both OCT and OCTA. Features were extracted from both OCT and OCTA. From OCT, the curvature, reflectivity, and thickness of retinal layers were extracted. From OCTA, the area of the foveal avascular zone, the vascular caliber, the vessel density, and the number of bifurcation points were extracted. A random forest classifier was used for classification. Table 1 summarizes the traditional ML methods for early detection, diagnosis, and grading of DR.

**Table 1 bioengineering-09-00366-t001:** Traditional ML methods for early detection, diagnosis, and grading of DR.

Study	Goal	Features	Classifier	Database Size	Performance
Welikala et al. [135], 2015	Detection of new vessels from fundus images as an indication of PDR	Local morphology features + genetic feature selection algorithm	SVM	60 Images from MESSIDOR [153] and local Hospital	Sen = 1000 Spec = 0.975 per image
Prasad et al. [136], 2015	Detection of DR (two classes: non DR vs. DR) using fundus images	41-statistical and texture features+ Haar wavelet transform for feature selection + PCA for feature reduction	Back propagation neural network and one rule classifier	89 images from DIARETDB1 [154]	ACC = 93.8% for back propagation neural network and ACC = 97.75% for one rule classifier
Mahendran et al. [137], 2015	Classification of the data into normal vs. abnormal followed by classification of abnormal into moderate NPDR or severe NPDR using fundus images	Statistical and texture features using GLCM extracted from segmented images	SVM and neural network	1200 images from MESSIDOR database	ACC = 97.8% (SVM) and ACC = 94.7%, (neural network)
Bhatkar et al. [138], 2015	Detect DR using fundus images	Discrete Cosine transform and statistical features	Multi-layer perceptron neural network	130 images from DIARETDB0 database	Spef = 100% Sens = 100%
Labhade et al. [139], 2016	Classification of the data into four classes: normal, mild NPDR, severe NPDR, and PDR using fundus images	40 statistical and GLCM texture features	SVM, random forests, gradient boost, AdaBoost, Gaussian naive Bayes	1200 images from MESSIDOR database	Best ACC = 88.71 using SVM
Rahim et al. [140], 2016	Classification of the data into five classes: no DR, mild NPDR, moderate NPDR, severe NPDR, and PDR using fundus images	Three features (area, mean, and standard deviation) of two extracted regions using fuzzy techniques (retina and exudates)	SVM with RBF kernel	600 images from 300 patients collected at the Hospital Melaka, Malaysia	*ACC* = 93%, *Spef* = 93.62%, and *Sen* = 92.45%
Islam et al. [141], 2017	Discriminate between normal and DR using fundus images	Speeded up robust features	k-means, a bag of words approach, and SVM	180 fundus images	*ACC* = 94.4%, *Pre* = 94%, *F*1 = 94% *AUC* = 95%
Carrera et al. [142], 2017	Classifying nonproliferative DR into 4 grades using fundus images	Extract features from isolates blood vessels, microaneurysms, and hard exudates	SVM	400 images	Sen = 95%
Somasundaram and Alli [143], 2017	Differentiate between NPDR and PDR	Extraction of the candidate objects (blood vessels, optic nerve, neural tissue, neuroretinal rim, optic disc size, thickness and variance)	Bagging ensemble classifier	89 colors fundus images	ACC = 49% for DR detection
Eladawi et al. [149]	Detecting early DR using OCTA	Density, appearance of the retinal blood vessels, and distance map of the foveal avascular zone	SVM	105 subjects	ACC = 97.3%
Costa et al. [144]	Grading DR using fundus images	Joint optimization of the instance encoding and the image classification stages	Weakly supervised multiple instance learning framework	1200 (Messidor) 1077 (DR1) 5320 (DR2) images	AUC = 90% (Messidor) AUC = 93 % (DR1) AUC = 96% (DR2)
Alam et al. [150]	Early detection of DR using OCTA images	Blood vessel tortuosity, blood vascular caliber, vessel perimeter index, blood vessel density, foveal avascular zone area, and foveal avascular zone contour irregularity	SVM	120 images	AUC = 94.41 % (control vs. disease) AUC = 92.96% (control vs. mild)
Sandhu et al. [152], 2020	Diagnosis of NPDR using OCT and OCTA	Curvature, reflectivity, and thickness of retinal layers (OCT), Area of foveal avascular zone, vascular caliber, vessel density, and number of bifurcation points (OCTA)	Random forest	111 patients	ACC = 96%, Sen = 100%, Spec = 94%, AUC = 0.96 (OCT + OCTA)
Sharafeldeen et al. [145], 2021	Detecting DR using OCT	Thickness, tortuosity, and reflectivity of 12 extracted retinal layers	Two-level neural networks	260 images from 130 patients	Sen = 96.15%, Spef = 99.23% *F*1 = 97.66% AUC = 97.69%
Liu et al. [151], 2021	Detecting DR using OCTA	A discrete wavelet transform was applied to extract texture features from each image	Logistic regression, logistic regression regularized with the elastic net penalty, SVM, and the gradient boosting tree	114 DR images + 132 control images	ACC = 82% AUC = 0.84 (logistic regression)
Wang et al. [146], 2021	Grading DR using OCT images	Foveal avascular zone (FAZ) metrics, Vessel density, extrafoveal avascular area and vessel morphology metrics	Multivariate regression analysis was used to identify the most discriminative features	105 eyes from 105 patients	Sen = 83.72% Spef = 78.38%
Abdelsalam et al. [147], 2021	Diagnosis of early NPDR using OCTA	Multifractal geometry	SVM	170 eye images	ACC = 98.5%, Sens = 100%, Spef = 97.3%
Elsharkawy et al. [148], 2022	Detection of DR using OCT	Gibbs energy extracted from 12 retinal layers	Majority voting using an ensemble of Neural networks	188 3D-OCT subjects	ACC = 90.56% (4-fold cross validation)

### 5.2. Deep Learning Methods

The most popular deep learning method is the CNN, which is composed of two types of layer: convolutional layers and fully connected layers. Convolutional layers are used to extract low- and high-level compact features, whereas the fully connected layers are used for classification. Different deep learning features may be applied, including transfer learning, data augmentation, and ensemble learning. Figure 6 summarizes the deep learning methods used for the detection, diagnosis, and grading of DR.

For fundus images, different deep learning methods have been applied. For example, Gulshan et al. [155] used an ensemble of 10 CNN networks for the grading of DR and DME using fundus images. The final decision of the ensemble was computed as the linear average of the predictions of the ensemble. Colas et al. [156] graded DR using a deep CNN network applied on fundus images. Their technique provided the location of the detected anomalies. Ghosh et al. [25] applied data augmentation, normalization, and denoising preprocessing stages. Then, a 28-layer CNN was applied for the grading of DR using fundus images. Takahashi et al. [157] differentiated between NPDR, severe NPDR, and PDR using fundus images. They applied a modified GoogleNet on the fundus scans to perform both feature extraction and classification. An ensemble of 26-layer ConvNets was used by Quellec et al. [158] for the grading of DR using fundus images. Ting et al. [134] identified DR and related eye diseases using an adapted VGGNet architecture. An ensemble of two networks was used for the detection of referable DR from fundus images. A zoom-in network was applied by Wang et al. [159] for the diagnosis of DR and the identification of suspicious regions using fundus images. Dutta et al. [160] compared back propagation NN, Deep NN, and VGG16-based CNN for the differentiation between mild NPDR, moderate NPDR, severe NPDR, and PDR using fundus images. The deep NN achieved the best performance. Zhang et al. [161] diagnosed the severity of DR using DR-Net with adaptive cross-entropy loss. Chakrabarty et al. [162] resized the grey-level fundus scans and input them to a nine-layer CNN in order to detect DR. Kwasigroch et al. [163] input the fundus images into a VGGNet in order to detect and stage DR. Li et al. [164] enhanced the contrast of fundus scans and input them into a transfer learning Inception-v3 CNN in order to detect referral DR. Nagasawa et al. [165] differentiated between nonPDR and PDR using ultrawide-field fundus images. Transfer learning of Inception-v3 CNN was used. Metan et al. [166] used ResNet for DR staging using color fundus images. Qummar et al. [167] used an ensemble of five CNNs, i.e., ResNet50, Inception-v3, Xception, Dense121, and Dense 169 to perform DR staging using fundus images. Sayres et al. [168] used Inception-v4 for DR staging using fundus images. Sengupta et al. [169] applied data preprocessing steps followed by an Inception-v3 CNN for DR staging using fundus images. Hathwar et al. [41] used a transfer learning Xception method for DR detection and staging using fundus images. Narayanan et al. [170] detected and graded the fundus images by investigating transfer learning of different networks, including AlexNet, VGG16, ResNet, Inception-v3, NASNet, DenseNet, and GoogleNet. Shankar et al. [171] applied histogram-based segmentation to extract the details of the fundus image. Synergic deep learning was performed for DR grading using fundus images. He et al. [172] graded DR using fundus images. They used CABNet, which is an attention module with a global attention block. They used DenseNet-121 as a backbone network of CABNet. Saeed et al. [173] applied transfer learning using two pretrained CNNs for DR grading using fundus images. Wang et al. [174] applied transfer learning using two networks, i.e., Inception-v3 and lesionNet, for DR grading. Hsieh et al. [175] used VeriSee™ software, which is based on a modified Inception-v4 model as a backbone network to perform DR grading using fundus images. Khan et al. [176] graded DR using a VGG-NiN model, which is formed by stacking VGG16, a spatial pyramid pooling layer and network-in-network. Zia et al. [177] applied feature selection and fusion steps. Then, the use of a CNN, including VGGNet and Inception-v3 models, was investigated for DR grading. Das et al. [178] detected and classified DR using fundus images. A CNN is built in which the number of layers was selected using a genetic algorithm. an SVM was used for classification. For grading DR, Tsai et al. [179] applied transfer learning using three models, i.e., Inception-v3, ResNet101, and DenseNet121. 

Using FFA images, Gao et al. [180] graded DR by investigating three deep networks, i.e., VGG16, ResNet50, and DenseNet. VGG16 achieved the best performance, with an accuracy of 94.17%.

Using OCT images, Eltanboly et al. [181,182] detected and graded DR by extracting features, including the reflectivity, curvature, and thickness of twelve segmented retinal layers. Deep fusion of the features was performed using auto-encoders. Li et al. [183] applied a deep network, called OCTD_Net, for early detection of DR using OCT images. Ghazal et al. [184] developed early detection of NPDR using OCT images based on an AlexNet followed by an SVM for classification.

Using OCTA images, Heisler et al. [185] applied ensemble training based on majority voting or stacking techniques using four fine-tuned VGG19. A maximum accuracy of 92% was achieved using the majority voting techniques. Ryu et al. [186] used ResNet101 for early detection of DR using OCTA. Using both OCT and OCTA, Zang et al. [187] classified DR using a network called DcardNet. He achieved an ACC of 95.7% for the detection of referable DR. Table 2 summarizes the deep learning methods used for early detection, diagnosis, and grading of DR.

## 6. The Role of AI in the Early Detection, Diagnosis, and Grading of AMD

In developed countries, AMD is a common eye disease in elderly people. OCT and other imaging modalities are used to detect and diagnose AMD. Subjective diagnosis is tedious and depends on the operator. With the invention of machine and deep learning, systems for the early detection, diagnosis, and grading of AMD have been designed to aid radiologists. In this section, we will briefly provide an overview of these methods. 

### 6.1. Traditional ML Methods

Different traditional ML methods based on image processing have been used for the early detection, diagnosis, and grading of ADM. Using color fundus images, García-Floriano et al. [190] differentiated normal from AMD with drusen. The image contrast was enhanced, followed by two morphological operations. Subsequently, invariant momenta were extracted and fed to an SVM, achieving an ACC of 92% for the identification of AMD with drusen from normal images. 

Using OCT, Liu et al. [191] used an automated method to identify normal and three types of retinal diseases (macular hole, macular edema, and AMD). Each SD-OCT image was encoded using spatially distributed multiscale texture and shape features. Two SVM classifiers with a radial basis kernel were trained to identify the presence of normal macula and each of the three pathologies, separately. For AMD, they achieved an AUC of 0.941. Srinivasan et al. [192] used SD-OCT to identify normal and two retinal diseases: dry AMD and diabetic macular edema (DME). Features were extracted using multiscale histograms of the oriented gradient descriptor, and an SVM was used for classification. They achieved an ACC of 100% for the identification of cases with AMD. Fraccaro et al. [193] used OCT images to automatically diagnose AMD with the aid of patient features, such as patient age, gender, and clinical binary signs (i.e., the existence of soft drusen, retinal pigment epithelium, defects/pigment mottling, depigmentation area, subretinal hemorrhage, subretinal fluid, macula thickness, macular scar, and subretinal fibrosis). They used two types of classifier: white boxes (interpretable techniques, including logistic regression and decision tree) and black boxes (less interpretable techniques, including SVM, random forest, and AdaBoost). Both types of classifier performed well, with an AUC of 0.92 using random forest, logistic regression, and adabosoost, and an AUC of 0.9 for SVM and decision tree. Soft drusen and age were identified as the most discriminating variables. A summary of the traditional ML methods for AMD detection, diagnosis, and/or staging is presented on Figure 7 and Table 3.

### 6.2. Deep Learning Methods

More recently, deep learning methods have been used for the detection, diagnosis, and grading of AMD. Using OCT, Lee et al. [194] modified a VGG19 CNN by exchanging the last fully connected layer with a fully connected layer of two nodes to support binary classification (i.e., two-class problem). Based on their network, they were able to differentiate between normal and AMD cases with an AUC of 92.77%, with an ACC of 87.6%, Sen of 84.6% and Spec of 91.5%, at the level of each image. Burlina et al. [195] used color fundus images to differentiate no or early AMD from intermediate or advanced AMD. They built an AlexNet architecture using a database of over 130,000 images from 4613 patients. They achieved an ACC from 88.4% to 91.6% and an AUC of 0.94 to 0.96. Teder et al. used transfer learning and Inception-v3 to detect exudative AMD from normal subjects using SD-OCT. Hassan et al. [196] segmented nine retinal layers and used SegNet followd by an AlexNet for the diagnosis of three retinal diseases (i.e., macular edema, central serous choriorentopathy, and AMD) using OCT. Motozawa et al. [197] used two 18-layer CNNs: A model to distinguish AMD from normal followed by a model to distinguish AMD with from AMD without exudative changes using SD-OCT scans. Li et al. [198] distinguished between normal, AMD, and diabetic macular edema using OCT images and transfer learning of a VGG-16 model. 

Using color fundus images, Ting et al. [134] identified three retinal diseases: DR, glaucoma, and AMD. They used an ensemble of two networks for the classification of each eye disease based on an adapted VGGNet architecture. They used a validation dataset of 71.896 images from 14,880 patients, achieving a Sen of 93.2% and Spef of 88.7% for identifying AMD. Tan et al. [199] achieved early detection of AMD using fundus images by applying data augmentation and a14-layer CNN model. An et al. [193] built two classifiers, one to detect AMD from normal and the other to detect AMD with fluid from AMD without fluid. They used two VGG16 models: a model to distinguish AMD from normal followed by a model to distinguish AMD with from AMD without fluid. Hwang et al. [200] distinguished between normal, dry (drusen), active wet, and inactive wet AMD using a cloud computing website [67]. The website was built using ResNet50, Inception-v3, and VGG16 networks. Figure 8 and Table 4 summarize the deep learning tools used for AMD detection and diagnosis.

## 7. Discussion and Future Trends

Artificial intelligence (AI) has demonstrated proof-of-concept in medical fields such as radiology and pathology, which have stunning similarities to ophthalmology, as they are intensely embedded in diagnostic imaging, the leading application of AI in healthcare [203,204,205]. The rapid expansion of AI facilities and their broad application continue to expand technological boundaries [206].

In ophthalmology, deep learning has been applied to automated diagnosis, segmentation, data analysis, and outcome predictions [1]. Several recent studies have used deep learning to diagnose and segment features of AMD [199,207] and DR, performing comparably to human experts [208,209].

One of the vital AI-based applications in ophthalmology is OCT image assessment, as the noninvasive, standardized, and rapid visualization of retinal pathology by OCT holds potential for the application of AI-based analyses [206]. AI not only allows knowledge to be generated based on large, multidimensional datasets, it is also able to capture individual variability in disease and function more efficiently than traditional methods [210]. We can summarize the findings in this survey as follows:Currently, FFA is the gold standard for assessing retinal vasculature, the most affected part of the retina in the diabetic eye. For early detection of DR, OCTA can detect changes in the retinal vasculature before developing DR clinical features.FFA and OCT are the gold standards for wet nAMD diagnosis [7,8].Currently, FAF and OCT are the basic methods for diagnosing and monitoring dry AMD. NIA, FFA and OCTA can provide complementary data [24].OCT is used to identify and monitor AMD and its abnormalities, such as drusen deposits, pseudodrusen, subretinal fluid, RPE detachment, and choroid NV [23].Using different medical image modalities, AI components have demonstrated outstanding capabilities to provide assisting automated early detection, diagnosis, and staging of DR and AMD diseases.Traditional ML methods are different with respect to the imaging modality used, the features extracted, and the classifiers used. For DR detection, diagnosis, and staging, fundus imaging, OCT, and OCTA have been used in the literature. For AMD detection, diagnosis, and staging, fundus imaging, FFA, OCT and OCTA have been used.Deep learning methods (mainly CNNs) have recently been introduced for the automated detection, diagnosis, and staging of DR and AMD diseases, achieving improved performance and representing the state of the art for the upcoming years. For DR detection, diagnosis, and staging, fundus imaging, OCT, and OCTA have been used. For AMD detection, diagnosis, and staging, fundus imaging and OCT have been used.

The future holds advances in technology:Using mixed image modalities for the eye will provide more information about the pathology, diagnosis, and proper treatments.Automated image interpretation using AI will play a dominant role in the early detection, diagnosis, and staging of retinal diseases, especially DR and AMD.Mobile applications are emerging, and can provide a fast, mobile solution for the early detection and diagnosis of retinal diseases.Large data sets will be acquired and available online for users. Quantification of large datasets will help to find reliable solutions.Further investigation into the relationship between retinal function and structure are required.

## 8. Conclusions

The current paper provided an in depth overview of the ophthalmic imaging modalities and their different types and different technologies in order to detect, diagnose, classify, and stage different retinal diseases, and more specifically, DR and AMD. In addition, the role of AI systems was surveyed from 1995 to 2022. Overall, AI systems are capable of assisting clinicians and providing an automated tool for the early detection, diagnosis, classification, and grading of DR and AMD. In the future, AI-based mobile solutions will be available.

## Figures and Tables

**Figure 1 bioengineering-09-00366-f001:**
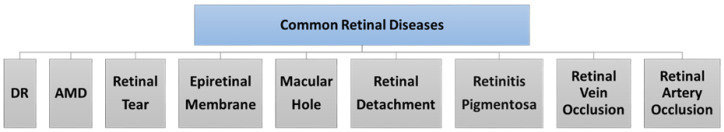
Common retinal diseases.

**Figure 2 bioengineering-09-00366-f002:**
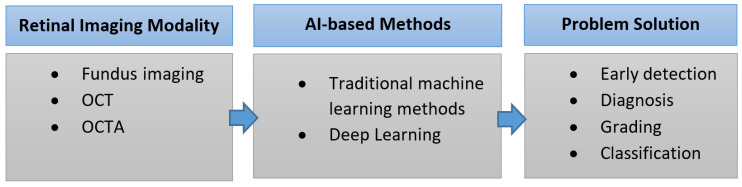
Analysis of retinal images.

**Figure 3 bioengineering-09-00366-f003:**
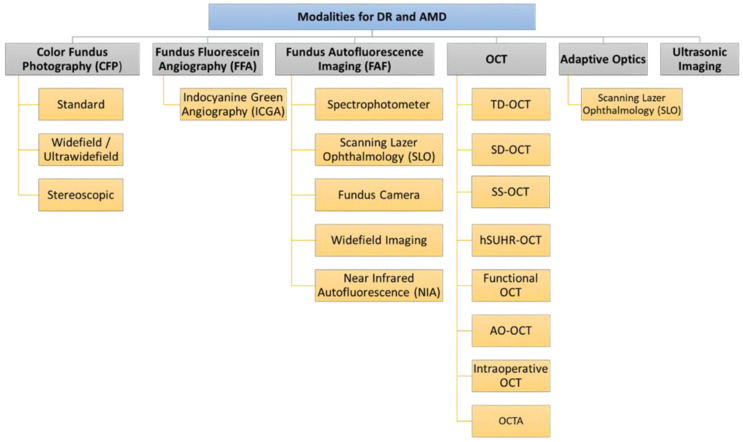
Medical image modalities for the detection, diagnosis, and staging of DR and AMD.

**Figure 4 bioengineering-09-00366-f004:**
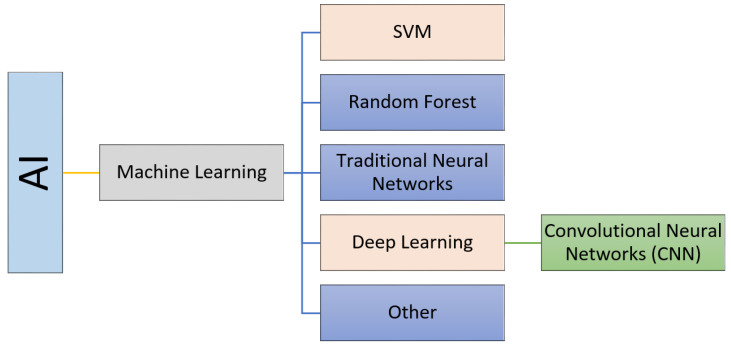
Components of artificial intelligence (AI).

**Figure 5 bioengineering-09-00366-f005:**
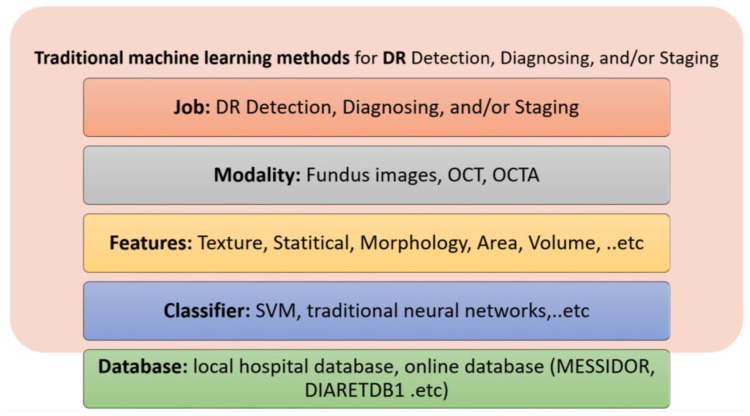
Summary of traditional ML methods for DR detection, diagnosis, and/or staging.

**Figure 6 bioengineering-09-00366-f006:**
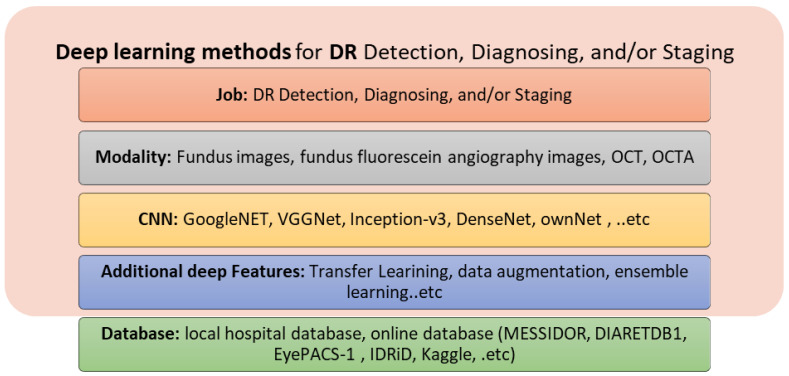
Summary of deep learning methods for DR detection, diagnosis, and/or staging.

**Figure 7 bioengineering-09-00366-f007:**
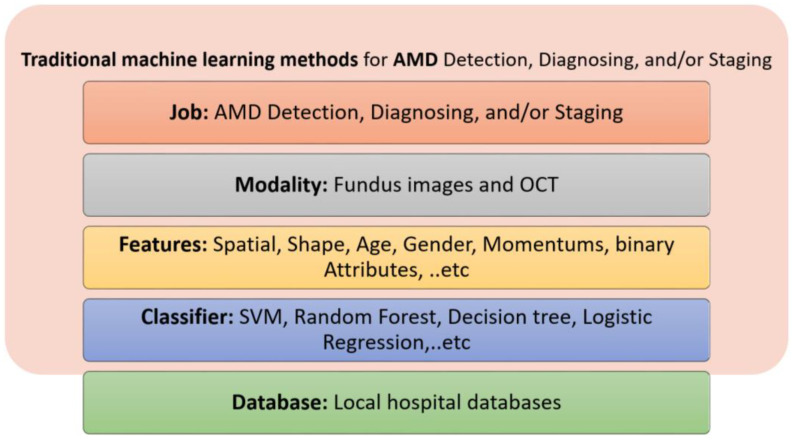
Summary of traditional ML methods for AMD detection, diagnosis, and/or staging.

**Figure 8 bioengineering-09-00366-f008:**
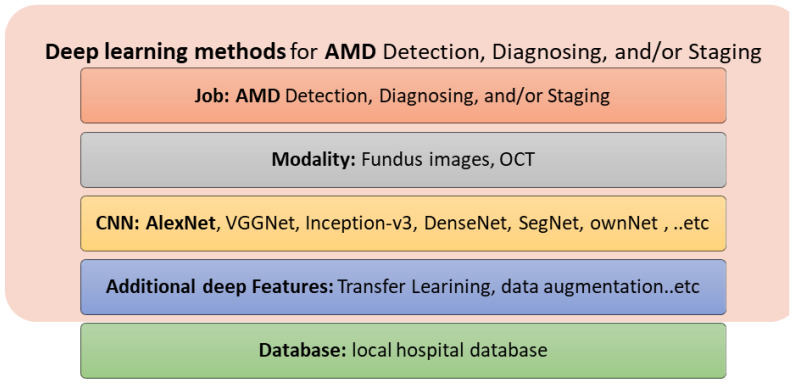
Summary of deep learning methods for AMD detection, diagnosis, and/or staging.

**Table 2 bioengineering-09-00366-t002:** Deep learning methods for early detection, diagnosis, and grading of DR.

Study	Goal	Deep Network	Other Features	Database Size	Performance
Gulshan et al. [155], 2016	Grading of DR and DME using fundus images	Ensemble of 10 CNN networks	Final decision was computed as the linear average of the predictions of the ensemble	128,175 + 9963 from EyePACS-1 +1748 from MESSIDOR-2	AUC = 99.1% (EyePACS-1) AUC = 99% (Messidor-2)
Colas et al. [156], 2016	Grading of DR using fundus images	Deep CNN network	Their technique provides the location of the detected anomalies	70,000 image (training) +10,000 (test)	*AUC* = 94.6%, *Sen* = 96.2%, *Spef* = 66.6%
Ghosh et al. [188], 2017	Grading of DR using fundus images	28-layer CNN	Data augmentation, normalization denoising were applied before the CNN	30,000 Kaggle images	*ACC* = 95% (two-class) *ACC* = 85% (five-class)
Eltanboly et al. [181], 2017	DR detection using OCT images	Deep fusion classifier using auto-encoders	Features are: reflectivity, curvature, and thickness of twelve segmented retinal layers	52 scans	*ACC* = 92% *Sen* = 83%, and *Spef* = 100%
Takahashi et al. [157], 2017	Differentiate between NPDR, Severe NPDR, and PDR using fundus images	Modified GoogleNet	Fundus scans are the inputs to the Modified GoogleNet	9939 scans from 2740 patients	ACC = 81%
Quellec et al. [158], 2017	Grading DR using fundus images	26-layer ConvNets	An ensemble of ConvNet was used	88,702 scans (Kaggle) +107,799 images (e-optha)	*AUC* = 0.954 (Kaggle) *AUC* = 0.949 (e-optha)
Ting et al. [134], 2017	Identifying DR and related eye diseases using fundus images	Adapted VGGNet architecture	An ensemble of two networks for detecting referable DR	494,661 images	*Sen* = 90.5% *Spef* = 91.6% for detecting referable DR
Wang et al. [159]	Diagnosing DR and identifying suspicious regions using fundus images	Zoom-in-Net	Inception-Resnet for the backbone network	35k/11k/43k for train/val/test (EyePACS) and 1.2k (Messidor)	*AUC* = 0.95 (Messidor) *AUC* = 0.92 (EyePACS)
Dutta et al. [160], 2018	Differentiate between mild NPDR, moderate NPDR, severe NPDR, and PDR	Back propagation NN, Deep NN, and CNN	CNN used VGG16 model	35,000 training and 15,000 test images (Kaggle)	*ACC* = 86.3% (DNN) *ACC* = 78.3% (VGGNet) *ACC* = 42% (back propagation NN)
Eltanboly et al. [182], 2018	Grading of nonproliferative DR using OCT images	Two-stage deep fusion classifier using autoencoder	Features are: reflectivity, curvature, and thickness of twelve segmented retinal layers	74 OCT images	*ACC* = 93% *Sen* = 91%, *Spef* = 97% (for detecting DR) *ACC* = 98% (for detecting early stage from mild/moderate DR)
Zhang et al. [161], 2018	Diagnose the severity of diabetic retinopathy (DR)	DR-Net with an adaptive cross-entropy loss	Data augmentation is applied	88,702 images from EyePACS dataset	ACC = 82.1%
Chakrabarty et al. [162], 2018	DR detection using fundus images	9-layer CNN	Resized grey-level Fundus scans are the inputs to the CNN	300 images	ACC = 100% Sen = 100%
Kwasigroch et al. [163], 2018	DR detection and staging using fundus images	VGGNet	Fundus scans are the inputs to the CNN	88,000 images	*ACC* = 82% (DR detection) *ACC* = 51% (DR staging)
Li et al. [164], 2019	Detection of referral DR using fundus images	Inception-v3	Enhanced contrast scans are the inputs to the CNN, Transfer learning is applied	19,233 images from 5278 patients	*ACC* = 93.49% *Sen* = 96.93% *Spef* = 93.45% *AUC* = 0.9905
Nagasawa et al. [165], 2019	Differentiate between nonPDR and PDR using ultrawide-field fundus images	Inception-v3	Transfer learning is applied	378 scans	*Sen* = 94.7% *Spec* = 97.2% *AUC* = 0.969
Metan et al. [166], 2019	DR staging using fundus images	ResNet	Color fundus images are the inputs to the CNN	88,702 (EyePacks)	ACC = 91%
Qummar et al. [167], 2019	DR staging using fundus images	Five CNNs: ResNet50, Inception-v3, Xception, Dense121, and Dense 169	Ensemble of five CNN	88,702 (EyePacks)	*ACC* = 80.80%, *Recall* = 51.50%, *Spef* = 86.72%, *F*1 = 53.74%
Sayres et al. [168], 2019	DR staging using fundus images	Inception-v4	Fundus images are the inputs to the CNN	1769 images from 1612 patients	ACC = 88.4%
Sengupta et al. [169], 2019	DR staging using fundus images	Inception-v3	Data preprocessing is applied	Kaggle EYEPACS and Messidor datasets	*Sen* = 90% *Spef* = 91.94% *ACC* = 90.4
Hathwar et al. [189], 2019	DR detection and staging using fundus images	Xception	Transfer learning is applied	35,124 images (EyePACS) 413 images (IDRiD)	Sen = 94.3% (DR detection)
Li et al. [183], 2019	Early detection of DR using OCT images	OCTD_Net	Data augmentation is applied	4168 OCT images	*ACC* = 92% *Spef* = 95% *Sen* = 92%
Heisler et al. [185], 2020	Classifying DR Using OCTA images	Four fine-tuned VGG19	Ensemble training is applied based on majority voting or stacking	463 volumes from 360 eyes	ACC = 92% (majority voting) ACC = 90% (stacking)
Zang et al. [187], 2020	Classifying DR Using OCT and OCTA images	DcardNet	Data augmentation is applied	303 eyes from 250 participants	ACC = 95.7% (detecting referable DR)
Ghazal et al. [184], 2020	Early detection of NPDR using OCT images	AlexNet	SVM was used for classification	52 subjects	ACC = 94%
Narayanan et al. [170], 2020	detect and grade the fundus images	AlexNet, VGG16, ResNet, Inception-v3, NASNet, DenseNet, GoogleNet	Transfer Learning is applied for each network	3661 images	ACC = 98.4% (detection) ACC = 96.3% (grading)
Shankar et al. [171], 2020	DR grading using fundus images	Synergic deep learning	Histogram-based segmentation was applied to extract the details of the fundus image	1200 images (MESSIDOR dataset)	*ACC* = 99.28%, *Sen* = 98%, *Spef* = 99%
Ryu et al. [186], 2021	Early detection of DR using OCTA	ResNet101	OCTA images are the inputs to the CNN	496 eyes	*ACC* = 91–98% *Sen* = 86–97%, *Spef* = 94–99%, *AUC* = 0.919–0.976.
He et al. [172], 2021	Grading DR using fundus images	CABNet with DenseNet-121 as a backbone network	CABNet is an attention module with global attention block	1200 images (MESSIDOR), 88,702 (EyePACS)	ACC = 93.1% AUC = 0.969 Per = 92.9%
Saeed et al. [173], 2021	Grading DR using fundus images	Two pretrained CNNs	Transfer Learning is applied	1200 images (MESSIDOR), 88,702 (EyePACS)	ACC = 99.73% AUC = 89% (EyePACS)
Wang et al. [174], 2021	Grading DR using fundus images	Inception-v3 + lesionNet	Transfer Learning is applied	12,252 images + 565 (external test set)	AUC = 94.3% Sen = 90.6% Spef = 80.7%
Hsieh et al. [175]	Grading DR using fundus images	VeriSee™ software	Modified Inception-v4 model as backbone network	7524 images	Sen = 92.2% Spec = 89.5% AUC = 0.955 (detecting DR)
Khan et al. [176]	Grading DR using fundus images	VGG-NiN model	VGG16, spatial pyramid pooling layer and network-in-network are stacked to form VGG-NiN model	25,810 images	AUC = 0.838
Gao et al. [180], 2022	Grading DR using fundus fluorescein angiography images	VGG16, ResNet50, DenseNet	Images are the inputs to the CNNs	11,214 images from 705 patients	ACC = 94.17% (VGG16)
Zia et al. [177], 2022	Grading DR using fundus images	VGGNet and Inception-v3	Applied a feature fusion and selection steps	35,126 Kaggle dataset	ACC = 96.4%
Das et al. [178], 2022	Detecting and classifying DR using fundus images	A CNN is used with several layers that is optimized using a genetic algorithm	SVM was used for classification	1200 images (Messidor dataset)	ACC = 98.67% AUC = 0.9933
Tsai et al. [179], 2022	Grading DR using fundus images	Inception-v3, ResNet101, and DenseNet121	Transfer Learning is applied	88,702 images (EyePACS) 4038 images	ACC = 84.64% (Kaggle) ACC = 83.80 (Taiwanese dataset)

**Table 3 bioengineering-09-00366-t003:** Traditional ML methods for early detection, diagnosis, and grading of AMD.

Study	Goal	Features	Classifier	Database Size	Performance
Liu et al. [191], 2011	Identify normal and three retinal diseases using OCT images: AMD, macular hole, and macular edema	Spatial and shape features	SVM	Train: 326 scans from 136 subject (193 eyes) Test:131 scans from 37 subjects (58 eyes)	AUC = 0.975; to identify AMD from normal subjects
Srinivasan et al. [192], 2014	Identify normal and two retinal diseases using SD-OCT: dry AMD and diabetic macular edema (DME)	Multiscale histograms of oriented gradient descriptors	SVM	45 subjects: 15 normal, 15 with dry AMD, and 15 with DME	ACC = 100% for identifying cases with AMD
Fraccaro et al. [193], 2015	To diagnose AMD using OCT images	Patient age, gender, and clinical binary attributes	White boxes (e.g., logistic regression & decision tree) and black boxes (e.g., SVM & random forest)	487 patients (912 eyes): 50 bootstrap test	AUC = 0.92
García-Floriano et al. [190], 2019	To differentiate normal from AMD with drusen using color fundus images	Invariant momentums extracted from contrast enhanced, morphological processed images	SVM	70 images: 37 healthy and 33 AMD with drusen	ACC = 92%

**Table 4 bioengineering-09-00366-t004:** Deep learning methods for early detection, diagnosis, and grading of AMD.

Study	Goal	CNN	Other Features	Database Size	Performance
Lee et al. [194], 2017	To differentiate between normal and AMD cases using OCT	Modified VGG19	A modified VGG19 DCNN with changing the last fully connected layer with a two-nodes layer	80,839 images for training and 20,163 images for test	*AUC* = 92.77%, *ACC* = 87.6%, *Sen* = 84.6% *Spef* = 91.5%
Ting et al. [134], 2017	Identify three retinal diseases: DR, glaucoma, AMD using color fundus images	Adapted VGGNet model	An ensemble of two networks is used for the classification of each eye disease	Validation dataset of 71,896 images; from 14,880 patients	Sen = 93.2% Spef = 88.7
Burlina et al. [195], 2017	Identify no or early AMD from intermediate or advanced AMD using fundus images	AlexNet	Solving two-class problem	130,000 images from 4613 patients	ACC = 88.4% to 91.6% AUC = 0.94 to 0.96
Treder et al. [201], 2018	Detect exudative AMD from normal subjects using SD-OCT	Inception-v3	Transfer learning	1012 SD-OCT scans	ACC = 96% Sen = 100% Spef = 92%
Tan et al. [199], 2018	Early detect AMD using fundus images	14-layer CNN model	Data augmentation	402 normal, 583 early, intermediate AMD, or GA, and 125 wet AMD eyes	ACC = 95% Sen = 96% Spef = 94% 10-fold cross-validation
Hassan et al. [196], 2018	Diagnosis of three retinal diseases (i.e., macular edema, central serous choriorentopathy, and AMD) using OCT	SegNet followed by an AlexNet	Segmenting nine retinal layers	41,921 retinal OCT scans for testing and 4992 for training	ACC = 96%
An et al. [202], 2019	Two classifiers: AMD vs. normal and AMD with fluid vs. AMD without fluid	Two VGG16 models	A model to distinguish AMD from normal followed by a model to distinguish AMD with from AMD without fluid	1234 training data and 391 test data	ACC = 99.2% AUC = 0.999 to identify AMD from normal. ACC = 95.1% AUC = 0.992 to distinguish AMD with from AMD without fluid
Motozawa et al. [197], 2019	Two classifiers: AMD vs. normal and AMD with exudative changes vs. AMD without exudative changes using SD-OCT images	Two 18-layer CNN	A model to distinguish AMD from normal followed by a model to distinguish AMD with from AMD without exudative changes	1621 images	ACC = 99% sen = 100% Spef = 91.8% to identify AMD from normal. ACC = 93.9% Sen = 98.4% Spef = 88.3% to identify AMD with from AMD without exudative changes
Hwang et al. [200], 2019	Distinguish between normal, Dry (drusen), active wet, and inactive wet AMD	ResNet50, Inception-v3, and VGG16	A cloud computing website [196] wasss developed based on their algorithm	35,900 images	ACC = 91.40% (VGG16), 92.67% (Inception-v3), and 90.73% (ResNet50)
Li et al. [198], 2019	Distinguish between normal, AMD, and diabetic macular edema using OCT images	VGG-16	Transfer learning	207,130 images	*ACC* = 98.6%, *Sen* = 97.8%, *Spef* = 99.4% *AUC* = 100%

## Data Availability

Not applicable.

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
