# Peer review of "The Role of Medical Image Modalities and AI in the Early Detection, Diagnosis and Grading of Retinal Diseases: A Survey"

_bioengineering, 2022, doi:10.3390/bioengineering9080366_

Round 1

Reviewer 1 Report

It is suggested to address the following comments:

1) Be careful not to replicate the same phrases in different sections of the manuscript. For example,  on lines 38 and 39 and the abstract.

2) It is necessary to have a small section or paragraph indicating the type of noise that can occur in this type of images, and some techniques suggested in
the literature to reduce its effect in later stages of detection, recognition and classification.

3) In relation to Figure 4, as well as its conceptualization, it is necessary to clarify the following: Artificial Intelligence is a field of knowledge (just like Biology or Chemistry, for example). Machine Learning is an important part of Artificial Intelligence but not the only one. Neural Networks are a popular type of Machine Learning, it is not the only one either. And Deep Learning are modern architectures of Neural Networks. In this sense, I argue that Artificial Intelligence is mainly divided into Machine Learning and Deep Learning, it seems counterproductive.  Since Deep Learning is part of Machine Learning.

4) In this research work it is necessary to mention image processing techniques, as well as characteristics generated or extracted from them. 
Since unlike Deep Learning, conventional Machine Learning techniques require features obtained from image processing.

5) Two tables are presented with a quantitative summary of methods to process this type of images. Discuss the results, suggest and support which one(s) could be considered the best  techniques and under what context. Since many times there are acceptable results with certain databases, but  performance is decimated by other sources of information.

Author Response

  • Be careful not to replicate the same phrases in different sections of the manuscript. For example, on lines 38 and 39 and the abstract.

Reply: Thank you very much. The replications were removed in the revised manuscript. Please, check lines 38 and 39 in the revised manuscript.

  • It is necessary to have a small section or paragraph indicating the type of noise that can occur in this type of images, and some techniques suggested in the literature to reduce its effect in later stages of detection, recognition, and classification.

Reply: Denoising represents a preprocessing step that helps to improve the AI efficiency for the detection, diagnosis, and staging of retinal diseases. Noise sources in CFP include additive and multiplicative noise [x1]. For OCT, noise sources include speckle noise, shot noise, and additive white Gaussian noise (AWGN) [x2]. For OCTA, noise sources include speckle noise and AWGN [x3]. For FFA, noise sources include the internal noise of sensitive components, optical material grain noise, thermal noise, transmission channel interference, and quantization noise [x4]. Most popular denoising techniques include using a Gaussian filter, a median filter, a wavelet filter, and/or a spatial domain filter. More recently, deep autoencoders play a significant role in image denoising. However, recent directions of deep learning attempt to efficiently train the deep learning network to process directly noisy images without the need of preproceesing techniques. In the revised version, Section 3 (Denoising of retinal images) has been added to detail denoising sources and methods, on lines 646-658 on page 14.

  • In relation to Figure 4, as well as its conceptualization, it is necessary to clarify the following: Artificial Intelligence is a field of knowledge (just like Biology or Chemistry, for example). Machine Learning is an important part of Artificial Intelligence but not the only one. Neural Networks are a popular type of Machine Learning, it is not the only one either. And Deep Learning are modern architectures of Neural Networks. In this sense, I argue that Artificial Intelligence is mainly divided into Machine Learning and Deep Learning, it seems counterproductive.  Since Deep Learning is part of Machine Learning.

Reply: Thank you very much for this comment. The reviewer is correct. Figure 4 is modified in the revised manuscript according to the reviewer's comment (DL is a part of ML). Following the reviewer's notice, we distinguished between two types of neural networks, traditional neural networks (neural networks that are composed of two layers; where the traditional backpropagation algorithm is used to adjust the weights) and deep neural networks (which are composed of many convolutional and fully connected layers). Please, see the changes in the revised manuscript (Fig. 4 on page 15, lines 660, 663, and lines 668-675 on page15)

  • In this research work it is necessary to mention image processing techniques, as well as characteristics generated or extracted from them. 
    Since unlike Deep Learning, conventional Machine Learning techniques require features obtained from image processing.

Reply: Thank you very much for this valuable comment. Traditional ML methods is based on image processing tools to extract the features from the images. In the updated manuscript, we emphasize of this. Please, see line 734 on page 17, lines 750, 751 on page 18, line 901 on page 26. The image processing techniques to extract image features are illustrated in detail on page 18, 19, and page 27 and Table 1 and 3 (in the features column). For example, on Table 1, the second reference, Prasad et al. used the Haar wavelet transform (an image processing technique) for feature selection and used PCA (an image processing technique) for feature reduction.

  • Two tables are presented with a quantitative summary of methods to process this type of images. Discuss the results, suggest and support which one(s) could be considered the best techniques and under what context. Since many times there are acceptable results with certain databases, but  performance is decimated by other sources of information.

Reply: 1.  More discussions and suggestions of the best techniques are included in the revised version, as follows:

  • Currently, FFA is the gold standard for assessing retinal vasculature, the most affected part of the retina in the diabetic eye. For early detection of DR, OCTA can detect changes in the retinal vasculature before developing DR clinical features.
  • FFA and OCT are the gold standards for wet nAMD diagnosis [7, 8].
  • Currently, FAF and OCT are the basic methods to diagnose and monitor dry AMD. NIA, FFA and OCTA can provide complementary data [24].
  • OCT is used to identify and monitor AMD and its abnormalities, such as drusen deposits, pseudodrusen, subretinal fluid, RPE detachment, and choroid NV [23].
  • Using different medical image modalities, AI components have proven outstanding capabilities to provide an assisting automated early detection, diagnosis and staging of DR and AMD diseases.
  • Traditional machine learning methods are different with respect to the imaging modality used, the feature extracted, and the classifier used. For DR detection, diagnosis, and staging, fundus imaging, OCT, and OCTA have been utilized. For AMD detection, diagnosis, and staging, fundus imaging FFA, OCT and OCTA have been utilized.
  • Deep learning methods (mainly CNNs) have been recently introduced for the automated detection, diagnosis, and staging of DR and AMD diseases, achieving improved performance and representing the state-of-the-art for the upcoming years. For DR detection, diagnosis, and staging, fundus imaging, OCT, and OCTA have been utilized. For AMD detection, diagnosis, and staging, fundus imaging and OCT have been utilized.

In the revised version, these findings has been added to section 6 (Discussions and future trends), lines 984-1005, on page 31.

[x1] Palanisamy, G., Shankar, N.B., Ponnusamy, P. and Gopi, V.P., 2020. A hybrid feature preservation technique based on luminosity and edge based contrast enhancement in color fundus images. Biocybernetics and Biomedical Engineering40(2), pp.752-763.

[x2] Chen, Q., de Sisternes, L., Leng, T. and Rubin, D.L., 2015. Application of improved homogeneity similarity-based denoising in optical coherence tomography retinal images. Journal of digital imaging28(3), pp.346-361.

[x3] Liu, H., Lin, S., Ye, C., Yu, D., Qin, J. and An, L., 2020. Using a dual-tree complex wavelet transform for denoising an optical coherence tomography angiography blood vessel image. OSA Continuum3(9), pp.2630-2645.

[x4] Cui, D., Liu, M., Hu, L., Liu, K., Guo, Y. and Jiao, Q., 2015. The Application of Wavelet-Domain Hidden Markov Tree Model in Diabetic Retinal Image Denoising. The open biomedical engineering journal9, p.194.

Special thanks to you for your good comments that really improve the quality and readability of the manuscript and solidify the concepts beyond the paper.

Reviewer 2 Report

In this review manuscript, the research progress on the variable imaging modalities for accurate diagnosis, early detection, and staging of both AMD and DR are summarized and reviewed. The recent reports of the AI technologies on the related diseases' automated detection, diagnosis, and staging are reviewed. This makes the manuscript suitable for "Machine Learning for Biomedical Applications"

The following issues should be addressed:

1. It is not suitable to say "The current paper is the first of its kind to overview in depth the ophthalmic imaging modalities and their different types and different technologies ...." in Conclusions. It is seldom to claim "the first" in a review paper. Review is a review, not a regular paper.

2.There are several typos in the manuscript. Please revise the typos.

Author Response

In this review manuscript, the research progress on the variable imaging modalities for accurate diagnosis, early detection, and staging of both AMD and DR are summarized and reviewed. The recent reports of the AI technologies on the related diseases' automated detection, diagnosis, and staging are reviewed. This makes the manuscript suitable for "Machine Learning for Biomedical Applications". 

The following issues should be addressed:

  1. It is not suitable to say "The current paper is the first of its kind to overview in depth the ophthalmic imaging modalities and their different types and different technologies ...." in Conclusions. It is seldom to claim "the first" in a review paper. Review is a review, not a regular paper.

Reply: Thank you very much for your comment. We remove this from the conclusion in the revised manuscript: line 1019 on page 31.

2.There are several typos in the manuscript. Please revise the typos.

Reply: We have made thorough and extensive editing and revisions of writing and formatting and conducted careful proofreading. The writing and readability of the paper have been much improved now. All changes are highlighted using track changes.

Special thanks to you for your good comments that really improve the quality and readability of the manuscript.

Reviewer 3 Report

The authors have done a great job in surveying over 200 papers concerning  DR detection  with the use of AI. I think the paper will be very usefull for the community. However, I think the way authirs see AI is slightly misleading. In fig. 4 you show machine learning and deep learning as separate areas, while in fact DL is just part of ML. As a result in subsequent figures you define neursl networks as classifier in ML and CNN in DL, but cnn stands for convolution nural networks soI think authors should tidy up a bit this approach to make things consistent.

Author Response

  1. The authors have done a great job in surveying over 200 papers concerning  DR detection  with the use of AI. I think the paper will be very usefull for the community.

Reply: Thank you for your positive attitude

  1. However, I think the way authirs see AI is slightly misleading. In fig. 4 you show machine learning and deep learning as separate areas, while in fact DL is just part of ML. As a result in subsequent figures you define neursl networks as classifier in ML and CNN in DL, but cnn stands for convolution nural networks soI think authors should tidy up a bit this approach to make things consistent.

Reply: Thank you very much for this comment. The reviewer is correct. Figure 4 is modified in the revised manuscript according to the reviewer's comment (DL is a part of ML). Following the reviewer's notice, we distinguished between two types of neural networks, traditional neural networks (neural networks that are composed of two layers; where the traditional backpropagation algorithm is used to adjust the weights) and deep neural networks (which are composed of many convolutional and fully connected layers). Please, see the changes in the revised manuscript (Fig. 4 on page 15, and lines 668-675 on page 15)

Special thanks to you for your good comments that really improve the quality of the manuscript and solidify the concepts beyond the paper.

Reviewer 4 Report

The authors introduced the present status of imaging technology on diabetic retinopathy and age-related macular degeneration and reviewed AI application on these two disorders. The contents of first half (introduction of imaging) are fair and almost reasonable from ophthalmological point of view. Tables 1-4 are key of this article and they look well reviewed.

I found some redundancy and duplication. For example, the paragraph from line 687 to 697 was duplicated. Please delete duplicated sentences and make it more compact.

Minor criticisms:

Line 590 ‘RPE’?   Type I CNV does no penetrate the RPE layer.

Line 708. Please spell out SVC, support-vector machine.

Line 849. ADM AMD

English edition is needed.

There are some typos.

Author Response

  1. The authors introduced the present status of imaging technology on diabetic retinopathy and age-related macular degeneration and reviewed AI application on these two disorders. The contents of first half (introduction of imaging) are fair and almost reasonable from ophthalmological point of view. Tables 1-4 are key of this article and they look well reviewed.

Reply: Thank you very much for your positive attitude

  1. I found some redundancy and duplication. For example, the paragraph from line 687 to 697 was duplicated. Please delete duplicated sentences and make it more compact.

Reply: Thank you very much. The indicated paragraph is removed from the revised manuscript: lines 716-730, pages 16 and 17

  1. Minor criticisms:

Line 590 ‘RPE’?   Type I CNV does no penetrate the RPE layer.

Reply: Thank you very much for this catch. We have corrected this in the revised version. Please see line 668, page 15 in the revised version.

Line 708. Please spell out SVC, support-vector machine.

Reply: Thank you. In the revised version, we have spelled out the SVM, line 655, page 14

Line 849. ADM → AMD

Reply: Thank you very much. In the revised version, we corrected this typo: line 893, page 26

  1. English edition is needed. There are some typos.

Reply: We have made thorough and extensive editing and revisions of English writing and formatting and conducted careful proofreading. The writing and readability of the paper have been much improved now. All changes are highlighted using track changes.